# Simplification of Administrative Procedure on the Example of the Czech Republic, Poland, Slovakia, and Hungary (V4 Countries)

**Lukáš Potěšil** [1,*] , **Krisztina Rozsnyai** [2] , **Jan Olszanowski** [3] **and Matej Horvat** [4]

1  Faculty of Law, Masaryk University, 611 80 Brno, Czech Republic
2  Faculty of Law, Eötvös Loránd University, H-1053 Budapest, Hungary; rozsnyaik@ajk.elte.hu
3  Faculty of Law and Administration, Adam Mickiewicz University, 61-712 Poznań, Poland; jan.olszanowski@amu.edu.pl
4  Faculty of Law, Comenius University in Bratislava, 810 00 Bratislava, Slovakia; matej.horvat@flaw.uniba.sk
*  Correspondence: Lukas.Potesil@law.muni.cz

**Abstract:** The article deals with the idea of simplification of administrative procedure on the example of legal regulation that can be found in Poland, Slovakia, the Czech Republic, and Hungary. This legal regulation comes from the same or similar evolution and legal conditions. General legal regulation of administrative procedure is represented by so called Code of Administrative Procedure. Existence of such code in all mentioned countries might be regarded as a first step towards simplification. Using research methods—dogmatic, normative, and, namely, comparative—the article examines concrete examples of simplification in mentioned countries that have similar approaches in solving this demand. This article mentions possible views (or addressees) on the need of simplifications as well as possible limits of this issue. In this sense, the protection of the public interest and protection of rights of individuals presents certain limitations to simplification. Legal regulation of administrative procedure is complicated. Although each legal regulation is in detail specific, we can find some common solutions in particular legal regulation of simplifications. Such results of this article might be useful (not only) for further comparison in European countries.

**Keywords:** administrative procedure; code of administrative procedure; simplifications; V4 countries; Poland; Slovakia; the Czech Republic; Hungary





## 1. Introduction

Administrative procedure is an important phenomenon of administrative law. Administrative procedure helps to implement administrative law. The implementation of many activities of individuals is conditioned by their assessment in administrative procedure. Administrative procedure can be encountered quite often. Usually on the basis of administrative procedure (and administrative decision) it is possible to, e.g., study at public schools, carry out a certain business, to build, or obtain a permit (to drive a motor vehicle, felling certain categories of trees). Administrative procedure establishes an obligation, e.g., pay a fine for an administrative offense or expropriate land.

Administrative procedure is a universal way of deciding on rights and obligations of persons in the field of public administration and administrative law. Administrative procedure is important not only for its participants and their legal relations. Through administrative procedure administrative bodies protect and promote the public interest and participate in the regulation of social relations. At present, we will probably not be able to administrate public affairs without administrative procedure and it will not be possible to completely replace it.

At first sight, administrative procedure may have similar features to proceedings before a (civil) court. The reason is that in both types of procedures decisions on rights and obligations are made. However, administrative procedure is different from court

proceeding. In administrative procedure it is not the independent court that decides, but the administrative body. Both the subject-matter and the nature of the rights are often different in the procedures (Merkl 1932) and application of several principles also differs. The focal point of the proceedings also differs. Court procedures are typically focused on events and deeds in the past, decide disputes in connection with existing legal relationships while administrative procedures yield at the future: they create, modify or terminate legal relationships. So administrative procedural legal relationships are—contrary to court procedure—established prior to material legal relationships. Of course, there are also different pieces of law applied: in civil court procedures a decision is made on private law relations primarily on the basis of private law. Administrative bodies decide on rights and obligations in the field of public administration and upon administrative law. Administrative procedure is mostly written and not public, while court proceeding is governed by the principle of publicity. Not only from that reason, administrative procedure should not be as complex, detailed and formalized as court proceedings.

Due to this, it is possible to ask a question whether the legal regulation of administrative procedure is clear, understandable, and predictable. Related to this question is whether it would not be possible to simplify the administrative procedure in any way. However, possible simplification of administrative procedure encounters possible limits.

The purpose of this paper is to point out the starting points for simplification of administrative procedure. The paper deals with whether it is possible to find any limits that could hinder the simplification of administrative procedure. The paper also mentions whether within the Central European area of the so-called Visegrad countries (V4) it is possible to find a certain unifying approach in the simplification of administrative procedure, as well as whether and how such simplification has taken place in Poland, the Czech Republic, Hungary, and Slovakia. The reason leading to a possible comparison of these countries is given partly by the common legal development, as well as the proximity of legal regulations and approaches to solving identical legal issues.

The paper first focuses on the approach and definition of administrative procedure, as provided by the theory and current legislation of Central European (i.e., V4) countries. Administrative procedure is based on common traditions. If the theory of V4 countries on matters of administrative procedure is similar, then we can assume the same on possible simplifications in administrative procedure. In the Section 3.1 we focus on why it is appropriate to consider the simplification of administrative procedure and whether administrative procedure is a suitable platform for simplification. The question is in whose favor the simplification of administrative procedure should be (Section 3.2). Further the paper focuses on the possible limits of simplification of administrative procedure and its possible limits (Section 3.3). Other part will explain how the legal regulation of the V4 countries approaches the simplification of administrative procedure and whether the approaches in the simplification of administrative procedure are similar (Section 3.4). In the discussion we deal with a summary of these issues and various approaches tackling their solution.

## 2. Methodology

In order to formulate a clear conclusion we will apply several methods of scientific research traditionally used in legal research. The right mixture of the methods will lead to achieve the goal of this article. The main goal is to discuss whether Codes of Administrative Procedures in V4 countries stipulate common basis for simplification of administrative procedure. Since there is a strong common historical background that has affected development of legal frameworks in all V4 countries, we will focus on whether this led to a proximity of legal regulations in area of simplification of administrative procedure.

The research consists of analyses of existing laws in V4 countries and of the professional literature and case law and have been conducted using different research methods, especially the dogmatic and comparative method. The dogmatic method has facilitated an

analysis and interpretation of the current law and the formulation of conclusions related to the problem being researched.

The method of analysis will be used to analyze approaches of V4 legislators in defining administrative procedures. This method will further be used to study views of legal scholars on subject of administrative procedure written in text books or monographs. Since the article mostly focuses on nowadays legislative approaches, the analysis will focus on wording of respective Code of Administrative Procedure as is actually in effect. If there is a need to explain several institutes from the historical point of view, then in order to explain its today's status a historical method will be used.

The synthesis method will be used along with comparison method. Based on analysis of administrative procedures, the article will focus on interrelationships between the administrative procedures in V4 countries. This approach will give the answer on whether there is a common understanding of what administrative procedure is. In order to achieve this answer, the article will compare legal regulation of all V4 countries.

In this way we will get a common background on approaches to administrative procedure in V4 countries that will give us a suitable knowledge board to answer the main research question on common basis for simplification of administrative procedure in V4 countries and whether there are any limits to simplification of administrative procedure. This last point of view will be tackled based on method of induction. The use of method of induction is justified as it will allow us to state general conclusions based on the evaluation of basic scientific data that arose from our analysis. Therefore, the method of induction will be used mainly in discussion and the concluding part of the article

With the application of comparative method, various legal systems have been analyzed together to identify the similarities and differences in terms of their effectiveness in providing legal protection, the quickness of the proceedings and the overall structure of the public system. The comparative method has helped to create diverse solutions for shared problems that are regulated differently by different legal systems.

## 3. Results of Simplifications in V4 Countries

### 3.1. Nature and Legal Framework of Administrative Procedure

Administrative procedure is a procedure of an administrative body, in cooperation with its participants, which decides on their rights and obligations. The result of administrative procedure is an administrative decision. Administrative procedure is always conducted on a very specific matter with a clearly defined circle of participants (Potěšil et al. 2020; Skulová et al. 2017; Vrabko et al. 2019).

Administrative procedure is traditionally regulated in a procedural act, which is usually referred to as the "Administrative (Procedure) Code" [there are a number of possible terms used and associated abbreviations, such as CAP (Code of Administrative Procedure), GALA (General Administrative Law Act), APA (Administrative Procedure Act)]. The Administrative Procedure Code has the nature of a lex generalis. The Administrative Procedure Code generally regulates administrative procedure that is universally applicable. Individual special laws (lex specialis) may introduce a more or less deviating regime from the general regulation in the Administrative Procedure Code and for various administrative procedures. This practice is observed (not only) in Poland where more than 200 statutes modify the general course of administrative proceeding (Piątek 2017).

Until the 1920s, administrative procedure was not regulated in a general and unified form in an Administrative Procedural Code. On the contrary, various provisions were scattered in the regulations, some of which were not even of the nature of legal regulations. This unsatisfactory situation has been the subject of frequent criticism since the beginning of the allocation of competences to administrative authorities in the 19th century (Čížek 1888; Pražák 1905). Literature has pointed out that it is difficult to know a number of regulations for which it is often unknown whether they remain valid. This concerned in particular the period before and after the disintegration of Austria–Hungary. At this time, administrative courts played a crucial role (Horáková and Tomoszková 2011). Their case

law has often replaced the absence of legislation and the absence of basic rules of procedure (Zumbini 2019). From the point of view of the examined countries, it was primarily the case law of the Administrative Court in Vienna (the so called "October Act" Act No. 36/1876 Coll., which had introduced administrative justice and had established this administrative court) for the Czech Republic, the Slovak Republic and partly for Poland (Olechowski 2018) and from 1897 on the Hungarian Royal Administrative Court in Budapest for Hungary (Rozsnyai 2018).

This historical excursion raises the question whether the existence of the general rules of administrative procedure is an advantage for administrative procedure and if it represents simplification. With regard to the requirements of the principle of legality, protection of the rights of the parties to the procedure, as well as predictability of law, an affirmative answer can be given that the Administrative Procedure Code is an advantage. Similarly, the Administrative Procedure Code and the general codification of administrative procedure contained therein are themselves a substantial simplification. General information about all the Administrative Procedure Codes of the V4 countries can be found in Table 1.

**Table 1.** Codes of Administrative Procedure of the V4 countries.

| Information About Codes / States | Adoption of Actual (First) Codification | Official/Original Number of Provisions | Definition of Administrative Procedure (Art.) | Similar Definition of Administrative Procedure |
|---|---|---|---|---|
| Poland | 1960 (1928) | 269 | 1 | Yes |
| Slovakia | 1967 (1928) | 85 | 1(2) | Yes |
| Czech Republic | 2004 (1928) | 184 | 9 | Yes |
| Hungary | 2016 (1957) | 144 | 7(2) | Yes |

From this point of view attempts on acceptance of EU administrative procedure code roots from the same problems and may lead to the same solution, where the existence such general provision is an advantage (to this issue see later).

The Polish administrative procedure is traditionally understood as an organized sequence of procedural activities, which form an organized cycle aimed at achieving a specific goal of the procedure (Hauser and Piątek 2017). The main goal of administrative procedure is to issue a decision that will create the rights or obligations of the parties to these procedure, which may be also settled silently. The participants of these procedure mainly are the parties to the procedure, which is understood as entities whose legal interest or obligation relates to the subject of procedure. The procedure is usually two-instance. The Polish Code of Administrative Procedure ("Polish CAP"; Act from 14th of June 1960—Code of Administrative Procedure) has been amended many times and is now an extensive legal act with almost 300 articles. However, compared to the Polish Code of Civil Procedure or the Code of Criminal Procedure, it is considered as a synthetic act in the doctrine. A feature of the Polish legislature is its casuistic nature. This usually results in the spaciousness (or even verbosity) of legal act including codes.

The Slovak Administrative Procedure Code ["Slovak CAP"; Act No. 71/1967 Coll. on administrative procedure (Administrative Procedure Code) as amended]. Given the common statehood with the Czech Republic, this code stipulated administrative procedure in the Czech Republic too. This lasted up until 1 January 2006 when the new Administrative Procedure Code came into force in the Czech Republic defines administrative procedure in Article 1(1). Pursuant to this article, Slovak CAP applies to procedure in the field of public administration in which administrative bodies decide on the rights, interests protected by law or obligations of natural persons and legal persons unless a special act provides otherwise. The result of the procedure is a decision which changes the legal status of a person, i.e., it changes the range of rights, interests protected by law or obligations of the person (Košičiarová 2012). Unlike CAPs of other V4 countries, Slovak CAP is a set of general rules where all special acts stipulate exceptions to general rules (mainly competence of the administrative body, i.e., which administrative body will carry out the procedure).

Since Slovak CAP is a set of general rules within 85 articles, this Code is not casuistic in its form.

The Czech Administrative Procedure Code ("Czech CAP"; Act No. 500/2004 Coll., entered into force on 1 January 2006) defines administrative procedure in Article 9. It follows that administrative procedure consists of authoritative decisioning on the rights and obligations of individuals. The result of the administrative procedure is an administrative decision. Administrative procedure represents the core of the Czech CAP. The regulation of administrative procedure is comprehensive and covers 143 provisions, which is more than 75% of the total content of the Czech CAP. The subsidiarity of the Czech CAP follows from its Article 1(2) and is used in administrative procedure in relation to other special laws.

Contrary to this basic position, in Hungary the general rules are—as a Hungarian specificity—based on the principle of the primacy of general rules, due to which special laws may only derogate from the provisions of the general rules if permitted by the CAP. (György 2018). The clinging to this unrealistic principle finally resulted in a hollowed-out set of general rules backed up by numerous subsidiary rules. The newest code, Act No. CL of 2016 on the General Order of Administrative Procedure ("Hungarian CAP") contains 144 articles. The aim of this codification was to have a significantly shorter code than the previous one (Act No. CXL of 2004, with 174 articles). This was only formally achieved as reduction is due to omitting some guarantees and by transposing regulation to separate codes (like Act No. CXXV of 2017 on administrative sanctions). The first CAP contained only 98 articles. The central notion of administrative case is substantially the same as in the first Hungarian CAP entered into force in 1958: "a case means the process in which the authority in making its decision, establishes the rights or obligations of the party, adjudicates his legal dispute, establishes his violation of rights, verifies a fact, status or data, or operates a register, as well as enforces decisions concerning these."

The Administrative Procedure Codes of the V4 countries, in accordance with theory, define administrative procedure similarly. At the latest they were adopted in the 1960s and remained (with amendments—Polish CAP and Slovak CAP) or were replaced by the new regulation mostly after 2004 (Czech CAP and Hungarian CAP). The level of detail of administrative procedure contained in the CAP also coincides. With the exception of the Slovak CAP, the CAP of the V4 countries are relatively extensive and casuistic. Moreover, this leads to the length of the administrative procedure and to these negative consequences. From that reason is fully justified the idea of possible simplification of administrative procedure. On the other hand, unfortunately the theory of administrative law of the V4 countries does not deal with the issue of possible simplifications in a very detailed way.

The question of simplification is raised from a somewhat different angle in the European Union, where the codification of the general rules of administrative procedure is not achieved yet, whereas is the V4 the general codes may look back on a quite respectable past of more than half a decade. However, the problem of simplification of administrative procedure is also crucial from the European perspective, within the efforts for the codification of general rules of administrative procedure. This is especially important as all V4 countries are EU members. Research on codifying the administrative procedure was conducted primarily within the Research Network on EU Administrative Law (ReNEUAL), which developed its Model Rules as specific prospective legislative proposals. Meanwhile the European Parliament adapted the resolution of 9 June 2016 for an open, efficient, and independent European Union administration (2016/261 (C 86/126)), coming from the resolution of 15 January 2013 containing recommendations to the Commission on a Law of Administrative Procedure of the European Union. This explicitly states that properly structured and consistent administrative procedures support both an efficient administration and a proper enforcement of the right to good administration guaranteed as a general principle of Union law and under Article 41 of the EU Charter of Fundamental Rights. Such codification could help to simplify the legal system, enhance legal certainty, and fill gaps in the legal system (Hofmann et al. 2014). It should be noted that such EU legal regulation will not directly influence the legal regulation

of the Member States, however we can find opinion taking account broader interpretation of crucial Article 298 TFEU (Hofmann et al. 2014).

The resolution is the outcome of a long process of the European Parliament continuously urging the European Commission to come forward with a proposal for a codified administrative procedure. Up to now, European administrative procedure is in a phase all the V4 countries run through before. General rules exist on the one hand on differently developed levels in sectoral law (e.g., the Customs Code or the Council Regulation (EU) 2015/1589 of 13 July 2015) and on the other hand regarding single institutions of administrative procedure, e.g., Regulation (EEC, EURATOM) No 1182/71 of the Council of 3 June 1971 determining the rules applicable to periods, dates and time limits (Rozsnyai 2017).

The Code of Good Administrative Behavior should also be mentioned. This is an act of EU soft law, initially created by the European Ombudsman, and then adopted on 6th of September 2001 by the European Parliament (2011/C 285/03). It regulates some questions of administrative procedure as the obligations of civil servants.

Other European level might be represented by the Council of Europe and its soft-law contained in different recommendations. Toward the idea of simplification, is the importance of good governance (CM/Rec(2007)7). From this, but also from other documents, is the importance of the requirement of speediness (Article 7), which is also result of possible simplification.

### 3.2. Simplification of Administrative Procedures—Cui Bono?

Almost 100 years have passed since the first codifications and drafts for codification of administrative procedure in the V4 countries were accepted. Since that time, the legal regulation of administrative procedure since that time has become more extensive and is relatively detailed.

Administrative procedure is a procedure that takes into account the participation of participants and is applied very often. This leads to legitimate considerations as to whether it would not be appropriate to simplify the administrative procedure.

In view of the fact that the administrative procedure meets a wide range of persons, it is necessary to make a request that the administrative procedure be arranged clearly and that its legal regulation is comprehensible (not only) for the participants.

The simplification of administrative procedure can also be achieved by making its legal regulation more "transparent". Speed and economy of the procedure can be associated with the clarity and comprehensibility of the legal regulation of administrative procedure. From the point of view of the administrative procedure and its possible simplification, it should be emphasized that the aim of the procedure is not to conduct the administrative procedure itself, but to issue a decision.

First of all, the question is whether the administrative procedure can be simplified without changing its legislation in any way. Simplification would then consist in changing the current habits, practices, and attitudes of administrative bodies as well as participants. In case of administrative bodies, this option is possible. It would be associated with the need to make organizational and personnel changes, as well as a series of training and general education. An easy simplification could be when officials will be fully aware that they conduct administrative procedure in which its participants are often waiting for an administrative decision. It is also important that the officials communicate sufficiently with the participants in such a way that they understand each other. The issue of misunderstanding was dealt with by the Czech Supreme Administrative Court (in its judgment of 11 September 2008, file no. No. 1 As 30/2008, No. 1746/2009 Coll. NSS.), the Supreme Administrative Court stated that the "addressees" of public administration are for the most part legal laymen, who cannot be required to formulate their applications quite precisely, and name things with exact legal terms, or even cite precise legal provisions in applications. In the exercise of public power, administrative authorities must accept the use of common non-professional language by users of public administration. If the terms of common language are insufficient, giving rise to legal ambiguity from the point of view

of the administrative authority, the administrative authority must invite the applicant to specify the content of the application and explain why clarification is necessary. ".

Another question is whether administrative procedure can be simplified without any changes in legal regulation of the administrative justice. Administrative justice subsequently reviews administrative procedure and the issued administrative decision. It follows from international (Article 6 (1) of the Convention for the Protection of Human Rights and Fundamental Freedoms, or the Recommendation of the Committee of Ministers of the Council of Europe (20) 2004 on judicial review of administrative acts) and constitutional (See Article 36 (2) of the Czech Charter of Fundamental Rights and Freedoms; Article 184 of the Polish Constitution from 2nd April 1997; Article XXVIII. (7) of the Basic Law of Hungary and Article 46 (2) of Constitution of Slovak Republic) requirements that administrative procedure subject to subsequent review by independent courts through the issued administrative decision. It is quite evident that proceedings before administrative courts affect administrative procedure. First, by the existence of case-law and the requirements for administrative procedure and administrative bodies expressed in it, but also by the fact that administrative courts may annul administrative decisions and return cases back (with the binding opinion of the administrative court) to administrative bodies for a repeated administrative procedure. Ideally, changes in administrative procedure should be combined with changes in proceedings before administrative courts. Such simplification is not very valuable, as the administrative procedure will be fast, but the subsequent judicial review in administrative justice will take many years, which is the reality in the V4 countries.

Simplification could be done comprehensively or in the form of simplifying alternative solutions and approaches to administrative procedure. These simplifications would be applicable in specific cases.

In the case of simplification of administrative procedure, different views and expectations are given. These may come from participants in administrative procedure, from administrative authorities, from administrative courts and from the public.

From the point of view of a participant in administrative procedure, administrative procedure is not so important as the administrative decision that results from it. In this case, the simplification of the administrative procedure would involve it being as quick as possible and only as burdensome as necessary/the least oppressive for the party. On the other hand though, a very important function of administrative procedure is to grant legal protection of the rights and interests of the parties affected (Pitschas 1990, p. 110.). This is most evident in procedures conducted ex officio, mostly to impose obligations and sanctions, as well as in procedures where counter-interested parties or the affected public is taking part. In such constellations, the views on the simplification of administrative procedure of administrative bodies, as well as of parties who apply for permissions may be influenced by the idea that it is the participant who "complicates" the procedure and intentions by exercising his or her procedural rights. The possible complexity of the administrative procedure increases the possibility of obstructions on the part of the participants in the procedure and their representatives (Potěšil et al. 2019). Quite often, administrative procedure is conducted by persons without legal or comparable education, and these obstructions are a great source of frustration for these civil servants, so administrative bodies would simplify the administrative procedure as much as possible. From this point of view there are more expectations toward positive outcomes of simplification.

Administrative justice, which reviews administrative procedure and administrative decisions, cannot be overlooked in this regard. The administrative justice can balance the often conflicting views of the administrative procedure that the participants and the administrative bodies have. Unfortunately, administrative procedure is becoming more judicial and more formal. The courts have repeatedly called on the administrative authorities to record all the facts carefully and for their individual procedural steps to be carefully substantiated. In practice, this often means that the administrative decision is not written in a way that is intended for the participants, but in such a way as to satisfy the

requirements of the supervising body or the administrative court, which of course requires time for its proper formulation.

It is also interesting to view the simplification of administrative procedure through the lens of the public. Administrative procedure is traditionally governed by the principle of non-publicity (Skulová et al. 2017; Vrabko et al. 2019). The public aspect would probably not be an element of simplification, but vice versa. The public may show distrust in the decisions taken, especially if they are not properly communicated and justified. In this respect, the public's view on the simplification of administrative procedure could be that the administrative procedure is becoming more transparent, its outcome predictable and clearly explained.

The aspect of simplification of administrative procedure from the point of view of the public is important, e.g., in matters of environmental protection. Rules and conditions for the public (in matters of environmental protection, the public is called "interested public") to participate in administrative procedure must be clear. This arises from Convention on Access to Information, Public Participation in Decision-Making, and Access to Justice in Environmental Matters (Aarhus Convention). Pursuant to its Article 1, in order to contribute to the protection of the right of every person of present and future generations to live in an environment adequate to his or her health and well-being, each Party shall guarantee the rights of access to information, public participation in decision-making, and access to justice in environmental matters in accordance with the provisions of this Convention.

In order for a fluent administrative procedure that has impact on the environment, a state has to ensure transparent and easy measurements for the interested public to participate in such proceedings. Otherwise the proceeding would be prolonged based on unnecessary judicial actions filed by the interested public. Given the general outline of this article, we will discuss whether the legal regulation on interested public and their rights to participate in administrative proceedings and subsequent judicial proceedings can or cannot contribute to simplification of administrative procedure in other paper.

### 3.3. Possible Limits of Simplification of the Administrative Procedure

The current form of administrative procedure and its complexity is largely due to the fact that administrative procedure is based on a relatively high standard of protection of the rights of individuals. Any simplification should not be construed as a resignation or abandonment of this standard. The warning for possible simplifications is the risk of seemingly simple solutions. Any change, including simplification, takes some time to take effect in practice. Nothing will change in a short time.

The Codes of Administrative Procedure have a framework character, which ensure the possibility of adjusting the single procedure to its nature and subject. That may be the reason for difficulties in amendments focused on general simplifications of procedure.

There are several non-legal obstacles that affect simplification of administrative procedure. At the first place, there are technical obstacles which may impede simplification of the administrative procedure (administrative bodies keep administrative files in paper form and not in electronic form, therefore, the administrative procedure can seem archaic and not easily accessible). The actions of administrative bodies in the pandemic clearly unveiled this obstacle. On one hand, the parties to a procedure had limited access to the administrative files given their paper form and direct contact with the officials. On the other hand, the administrative bodies did not always have the possibility of informal communication with the parties, which could have sped up the administrative procedure. The second obstacle of a factual nature may be the officials' habituation to the course of procedural activities, their complexity, and formalism.

We can conclude that there are no legal limits that might prevent the simplification of administrative procedure. Public administration is conservative and critical of change. Furthermore, from the habit, some examples of simplification may not be used so much. These are also other possible limits that need to be taken into account.

### 3.4. Examples of Administrative Procedure Simplification in V4 Countries

As it was noted earlier, most of the changes introduced to the acts creating the shape of administrative procedure were aimed at increasing the efficiency of the procedure itself. One of the ways to achieve this goal is to speed up and simplify the procedure. The purpose of legal procedure (not only administrative, but also judicial) is to resolve an individual case in the shortest time possible while at the same time guaranteeing the result within a fair and legal process. Always when the changes have been introduced to the administrative procedure codes, the legislator had to weigh in two values: the right to obtain a fair decision and the right to hear the case without undue delay. The instruments introduced into the CPA were aimed at enabling the implementation of both of these demands. Examples of similarities or differences can be found in Table 2.

**Table 2.** Examples of Possible Simplifications in the Code of Administrative Procedure), (CAP) of the V4 Countries.

| Legal Instruments Leading to Simplification | Polish CAP | Slovakian CAP | Czech CAP | Hungarian CAP |
|---|---|---|---|---|
| The principle of speed and minimalization of interventions | Yes | Yes | Yes | Yes |
| Electronic delivery and delivery to data (electronic) boxes | Yes (on the demand of the party) | Yes | Yes | Partially, according to a special act |
| Delivery by the public notice | Yes | Yes | Yes | Yes |
| Procedure with a large number of participants | Yes | No | Yes | No |
| The possibility of waiving certain procedural rights (incl. appeal) | No | Yes | Yes | Yes |
| Simplified Decision (without reasoning) | Yes | Yes | Yes | Yes |
| Public law contract (that may replace decision) | No | No | Yes | No |
| Self-review of the first level decision | Yes | Yes | Yes | Yes |

The Polish CAP contains regulations which are referred to as general principles. Among these partially self-evident principles are also speed and simplicity. According to Article 12 (1) Polish CAP public administration authorities should deal with cases thoroughly and quickly, using the simplest available methods to resolve them. In the Polish CAP, the mechanisms enabling the simplification of the administrative procedure is also present. However, these are rather rights for the parties which can be used (i.e., the possibility of electronic service of letters, resignation from some procedural rights by the party). The simplification of the procedure imposed on the party by the law is much less frequent (i.e., delivery by the public notice in cases with a large number of participants, or the administration silent in some cases). In 2017 a special procedure allowing for simplified procedure was introduced (Art. 163b–163g Polish CAP), which allows for some simplifications during the procedure (i.e., limiting the number of parties to the procedure to the applicant only, possibility of submitting applications in the special form, rule of evidence preclusion, simplification of the justification of the decision, or limiting the range of orders issuing during the procedure that may be challenged in the course of the procedure).

In case of the Czech Republic, the legal regulation of administrative procedure is the opposite rather than a simplified one. At first sight, there has been an extreme increase in the general legal regulation of administrative procedure in the Czech CAP than in previous legal regulations (compared with the Slovak one). On the other hand, the case law of administrative justice imposes a large number of requirements, which make administrative procedure more complex and bureaucratic. This approach is also reflected in the legislation itself, which does not provide for simplistic approaches. However, in the current legislation, it is possible to find institutes for which simplification could take place. We can think about rights for the parties which can be used (i.e., the possibility of electronic delivery, or resignation from many procedural rights by the party, including appeal). The simplification of the procedure imposed on the party by the law is much less frequent (i.e., delivery by the public notice in cases with a large number of participants). There is no comprehensive

legal regulation to simplify the administrative procedure. However, as was mentioned, there are several institutions, which can be regarded as tools for simplifying administrative procedure. Majority of them (the simplified decision, as well as the possibility of the deciding authority to take back/alter its decision upon the appeal) were known before this legal regulation and are still used. The most important one—public law contract—that may replace administrative procedure and administrative decision is usually regarded as a dangerous tool due to its corruption threats. We can conclude that examples of simplifications are spread in the Czech CAP and do not differ from others examples. In addition to the Czech CAP, it is possible to find special legal regulations that introduce simplifying procedures in administrative procedure, such as e.g., (transport) infrastructure. The problem, however, is that these simplifications are based on shortening the time limits that can be encountered in the proceedings (e.g., for both decisions and appeals).

In Hungary, we can divide the question of the simplification of administrative procedure into several phases. This is due to the fact that public policy goals have changed significantly after 2010. We will thus handle the single eras separately.

Paradox as it may sound, the Hungarian CPA of 1957 was a compact Code with simple text, which also guaranteed numerous rights for the party. The reasons for this are twofold: on the one hand, most of the clerks working with the CPA did not have a legal education, nor another university degree, so the regulations had to be very plain and easy-to-understand. On the other hand, in the more important procedures (like on planting and issuing other permits for economic activities) the party was the state and its entities (state enterprises mostly), so the legislator was directly interested in creating simple and quick, but also party-friendly administrative procedures. The integration of procedures, the possibility of the waiver or withdrawal of appeal, the simplified decision, as well as the possibility of the deciding authority to take back/alter its decision upon the appeal were all institutions used already by this CPA made in one of the darkest times of socialism. So there was no real need for a new CPA, there were three decisions of the constitutional court upon which the problems regarding legal protection were settled easily. However, in 2004 a new code was codified, the adjustment in it to some European tendencies partly were tools of simplification, too. Here we can list the creation of a procedure for public participation, including the institution of the mediator for authoritative cases and the public consultation, as well as the institution of the authoritative contract. The law of the European Union also required the simplification of some institutions and procedures. The transposition of the Service Directive led to the introduction of one-stop-shops, the increasing possibility of silent decision-making (positive silence of administration) as well as to the replacement of permissions with duties to notify the authority (Rozsnyai 2008). The big planting procedures also obtained a special simplified regulation, mostly shorter time limits and shifting first instance decisions to supervisory authorities as well as omitting appeals.

After 2010, a new rhetoric made its way and in the course of the government program for the "reduction of overhead" (Kovács and Rozsnyai 2019), the cutting back of bureaucracy, not only the red tape, but also the administrative burdens of citizens was declared a major policy goal. This resulted in different institutions, from which we can regard the new threefold system of procedures as a simplification (Hoffman and Rozsnyai 2020). The real "simplification" was a heavy process of centralization (Fazekas et al. 2016) which led on the one hand to less administrative bodies, thus less need of integration of procedures and less inner-administrative communication, and on the other hand through the decrease of instances also to the abolition of the appellate procedure. The pandemic brought about a new wave of simplification and practically erased permissions and gave way to a so-called controlled notification in vast areas of public administration (Hoffman and Balázs 2020)—again an adjustment of procedural law to the scarcity of personnel communicated as a simplification.

Slovak CPA came into force almost 53 years ago (1 January 1968). Since then, only 11 amendments of this act has come into force with only one being a complex one. The rest of them only partially amended several provisions of CPA. Unlike Polish CPA, Slovak

CPA does not stipulate any special provisions on simplifications of administrative procedure. Slovak CPA stipulates only various legal institutes that tackle this issue. They are summarized in the Table 2 below.

The reason for the Slovak CPA to be still in force, even after 53 years, is that no Slovak government has ever identified itself with the idea of a new CPA. Given the fact that CPA's provisions are quite general, the legal practice does not indicate any motions that would call for adoption of a new legal regulation. Despite this fact, some 20 years ago, a draft of a new CPA was introduced, however this draft has never been officially introduced by any minister (as a member of the government) and it never made it to a form of a bill.

As mentioned, only several provisions of Slovak CPA tackle issue of simplification of administrative procedure. However, none of the provision look at the issue of simplification from a general perspective, they present rather partial views.

## 4. Discussion

On the question whether simplification is appropriate in the area of administrative procedure, we can conclude that yes. As was mentioned, administrative procedure is a frequent way for individuals to come into contact with the public administration. At the same time, they have expectations of rapid and as informal solution as possible. In practice, however, this is often not the case. In the V4 countries, one of the biggest problems is the length of administrative procedure.

If we asked whether the existence of the general legal regulation itself could already represent a certain simplification of the administrative procedure, we can answer positive here too. The past shows that the absence of unifying and general rules is a disadvantage for administrative procedure. It can be proven by the activities in the EU administrative procedural law and proposal of the EU Administrative Procedure.

Persons who conduct administrative procedure on behalf of an administrative body often do not have the necessary legal training. For (and not only) them is the existence of general legislation is an advantage. At the same time, however, this requires a proper understanding of the relationship lex specialis derogat lex generalis, as well as orientation in the relevant legislation.

Another question is what is the content of the Code of Administrative Procedure. The administrative procedure rules of the V4 countries, with the possible exception of Slovakia, are relatively extensive and detailed; although, they used to be shorter. However, this creates space for possible simplification of legislation, respectively adoption of simplification elements. As Hungarian changes show: the abolition of legal regulations does not lead to simplification but to even bigger complexity, as actors do not know to which rules they should adjust their acts and behavior—it only leads to legal uncertainty.

On the example of the administrative procedure of the V4 countries and examples of their possible simplification approaches, it is quite evident that the legal regulations are moving in a similar direction. This justifies the idea of comparison. The limitation of comparison comes from partial difference in administrative law of each V4 country. On the other hand our research has potential for the future to be inspired by different legal solution used in V4 countries and to prepare exhaustive idea of simplification.

In this paper, we looked at whether there are any limits that could be associated with the simplification of administrative procedure. We have come to the conclusion that any simplification of administrative procedure should always respect the requirement of protection of the public interest, as well as the achieved standard of protection of (procedural) rights in procedure. There are basically no other legal limits, so the only restrictions are rather factual in nature.

It is clear from the examples of simplification procedures of the V4 countries that they are primarily aimed at speeding up administrative procedure. We believe that ADR measures that are not based on traditional unilateral and sovereign practices that are otherwise typical of public administration could have some potential for simplification. In this respect, in particular, it is an institute of public law contracts, which may have this

nature. However, experience from their application in the Czech Republic shows a rather cautious approach.

With the exception of the Polish CPA, we will not find any comprehensive or complex approach to the issue of simplifying administrative procedure in the administrative regulations of the V4 countries, although the Polish CPA is not completely exhaustive. We can find many provisions in the individual administrative regulations that have simplifying potential, but mostly is made on shortening time limit.

We believe that an element of simplification could be changes in the perception of administrative procedure, abandoning legally formalistic and prudent approaches, without having to change the legislation in any way. Examples of the simplification of administrative procedure in the V4 countries, which are included in their administrative rules, consist mostly in the possibility of waiving a number of procedural rights and in the electronic method of delivery of different documents. In this, it is evident that the benefits of technological progress and the application of new and modern technologies can contribute to simplification, as the COVID-19 pandemic has shown in many fields.

An important element of simplification is the existence of a simplified decision, often without a detailed and comprehensive justification, or an element of "self-remedy", which, however, is not used very often.

We are of the opinion that the following facts will need to be taken into account in the eventual simplification of administrative procedure. First of all, it is a requirement for an "online" form. At the same time, the often-argued argument that "the law only delays" must be rejected. In addition, a thorough revision of hundreds of special laws as to whether and to what extent deviating from the Code of Administrative Procedure is still justified and can be considered an element of simplification.

Overall, we are of the opinion that the current legislation on simplification of administrative procedure is not sufficient. Within the framework of possible simplification approaches, the legislator should primarily focus on the use of e-government and new technologies and their application in administrative procedure.

**Author Contributions:** Although the manuscript is a result of a collaboration among the four authors, the contribution of each author can be qualified as follows: L.P. and K.R. wrote Section 1. Introduction; J.O. and M.H. wrote Section 2. Methodology. Section 3. Results of Simplifications in V4 Countries and Section 4. Discussion is a joint effort of the four authors. All authors have read and agreed to the published version of the manuscript.

**Funding:** This research was founded by VISEGRAD FUND, grant number 21910091.

**Institutional Review Board Statement:** Not applicable.

**Informed Consent Statement:** Not applicable.

**Data Availability Statement:** Not applicable.

**Conflicts of Interest:** The authors declare no conflict of interest.

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
