# Peer review of "Simplification of Administrative Procedure on the Example of the Czech Republic, Poland, Slovakia, and Hungary (V4 Countries)"

_admsci, doi:10.3390/admsci11010009_

Round 1

Reviewer 1 Report

The article analyzes the need to simplify tje administrative procedures with examples from the 4 states analyzed (Poland; Slovakia; the Czech Republic; Hungary) and has the merit of bringing to the fore the difficulties and limitations of the simplification and transparency process imposed by the approach to the real needs of citizens.

Author Response

We tried to make changes in English in the text.

The introduction contains some changes. We added differences between court and administrative procedure (line number 41 – 45), as well as the methodological part (line number 83 – 129). The methods are now adequately described and probably the introduction provides better background.

Reviewer 2 Report

Minor English changes required by professional English proofreader.

Author Response

We tried to make changes in English in the text.

Reviewer 3 Report

The manuscript is rather diagnostic - the author analyzes the simplification of administrative procedure on the example of the V4 countries. The topic raised in the manuscript is undoubtedly important, especially in the context of public governance, that the author points out to.

Unfortunately, this paper has deficiencies that I will describe below:

  1. The introduction must be substantially improved. The author provides very general information and vague ideas about the administrative procedure, in this section, it would be advisable to focus a bit more on European regulations and documentation about this issue, and prior works about the exposed problem. Also, there are some paragraphs that must be re-written: lines 22-35 present ideas disconnected by punctuation marks.
  2. The methodology is solved in one line in the introduction. I suggest writing a "methodology section" after the introduction in which the methodology used is explained in more detail and include a summary of the manuscript sections.
  3. The manuscript denotes some weaknesses in the theoretical support and I would suggest to improve Section 2. The theoretical framework begins with very abstract and generic ideas about the law, it would be necessary to make a critical discussion about the prior works and the European regulations/documentation on the administration and the need for simplification.
  4. In lines 391- 396 the examples of possible simplifications in the CAP are segregated among four tables by disseminating the findings instead of clearly expose the preliminary results obtained. I do highly recommend comprising this information by grouping up the variables.

Considering the previous comments, and although I recognize potential to the ongoing research, I cannot recommend that the paper be accepted in the present form.

Other minor suggestions:
1. What are the limitations of research?

2. What are the possibilities of future research?

3. Perhaps in the title of the paper, the author should not use "V4" and select Visegrad countries. Some readers might not understand it.

Author Response

We tried to make changes in English in the text.

Perhaps in the title of the paper, the author should not use "V4" and select Visegrad countries. Some readers might not understand it

We changed the title to be more understandable: Simplification of Administrative Procedure on the Example of the Czech Republic, Poland, Slovakia and Hungary (V4 Countries)

  1. The introduction must be substantially improved. The author provides very general information and vague ideas about the administrative procedure, in this section, it would be advisable to focus a bit more on European regulations and documentation about this issue, and prior works about the exposed problem. Also, there are some paragraphs that must be re-written: lines 22-35 present ideas disconnected by punctuation marks.
  2. The methodology is solved in one line in the introduction. I suggest writing a "methodology section" after the introduction in which the methodology used is explained in more detail and include a summary of the manuscript sections.

We added differences between court and administrative procedure (line number 41 – 45), as well as the methodological part (line number 83 – 129) in the introduction. The methods are now adequately described and probably the introduction provides better background. We did a short summary of the manuscript sections (line number 75 – 82). The European background in mentioned in chapter 2 (see below), even we think that European regulations and documentation are not focused on the issue of simplification itself.

3. The manuscript denotes some weaknesses in the theoretical support and I would suggest to improve Section 2. The theoretical framework begins with very abstract and generic ideas about the law, it would be necessary to make a critical discussion about the prior works and the European regulations/documentation on the administration and the need for simplification.

We added (line number 175 – 177 and namely 244 - 281) the wider European context, that is connected especially with the proposal/attempts of adoption of the EU Model Rules on Administrative Procedure.

4. In lines 391- 396 the examples of possible simplifications in the CAP are segregated among four tables by disseminating the findings instead of clearly expose the preliminary results obtained. I do highly recommend comprising this information by grouping up the variables.

We mixed all tables into one, so we hope that it would be more understandable.

Other minor suggestions:

  1. What are the limitations of research?
  2. What are the possibilities of future research?

We also put a short remark about limits and potential of our research (line number 558 – 561). 

Round 2

Reviewer 3 Report

The paper sounds now more focused and targeted at the specific aims of the author/s, that is the simplification of administrative procedure in V4 countries.

As asked, the author/s focused the introduction avoiding dispersion over generalities and centralizing on the specific aim and scope. Moreover, the methodology was also  improved and justified. However, to keep a clear structure, lines 87-109 should be included in a separate section: "2. Methodology".

The section of results was also updated and the author/s provided the European context. Finally, table two was mixed/compressed as I required. Nevertheless, there is an error, "table 1" appears when it should be "table 2".

As so, I see now that the manuscript can go further.

In this version, I have detected another error with the lines of table one (one of them is not centred). I'm not sure if it's intentional or a problem with the document.

Author Response

The paper sounds now more focused and targeted at the specific aims of the author/s, that is the simplification of administrative procedure in V4 countries.

As asked, the author/s focused the introduction avoiding dispersion over generalities and centralizing on the specific aim and scope. Moreover, the methodology was also  improved and justified. However, to keep a clear structure, lines 87-109 should be included in a separate section: "2. Methodology".

We made a separate section 2. Methodology, that includes now lines number 82 – 118.

The section of results was also updated and the author/s provided the European context. Finally, table two was mixed/compressed as I required. Nevertheless, there is an error, "table 1" appears when it should be "table 2".

The second table was changed as a „table 2“. It was a mistake. Now it is correct (number line 493)

In this version, I have detected another error with the lines of table one (one of them is not centred). I'm not sure if it's intentional or a problem with the document.

We made formal changes of both tables, so now it should be better and without errors (number lines 214).